# p38α MAPK Regulation of Energy Metabolism in Skeletal Muscle Offers a Therapeutic Path for Type 2 Diabetes

**DOI:** 10.3390/cells14161277

**Published:** 2025-08-18

**Authors:** Eyal Bengal, Sharon Aviram

**Affiliations:** Department of Biochemistry, The Ruth and Bruce Rappaport Faculty of Medicine, P.O. Box 9649, Bat Galim, Haifa 31096, Israel; avirams@technion.ac.il

**Keywords:** skeletal muscle, insulin resistance, p38α MAPK, AMPK, mitochondrial metabolism

## Abstract

Type 2 diabetes (T2D), a growing global health concern, is closely linked to obesity and sedentary behavior. Central to its development are insulin resistance and impaired glucose metabolism in peripheral tissues, particularly skeletal muscle, which plays a key role in energy expenditure, glucose uptake, and insulin sensitivity. Notably, increased accumulation of lipid metabolites in skeletal muscle is observed both in endurance exercise—associated with improved insulin sensitivity—and in high-fat diets that induce insulin resistance. The review examines the contrasting metabolic adaptations of skeletal muscle to these opposing conditions and highlights the key signaling molecules involved. The focus then shifts to the role of the stress kinase p38α mitogen-activated protein kinase (MAPK) in skeletal muscle adaptation to overnutrition and endurance exercise. p38α enhances mitochondrial oxidative capacity and regulates nutrient utilization, both critical for maintaining metabolic homeostasis. During exercise, it cooperates with AMP-activated protein kinase (AMPK) to boost glucose uptake and fatty acid oxidation, key mechanisms for improving insulin sensitivity. The co-activation of p38α and AMPK in skeletal muscle emerges as a promising therapeutic avenue to combat insulin resistance and T2D. The review explores strategies for selectively enhancing p38α activity in skeletal muscle. In conclusion, it advocates a comprehensive approach to T2D prevention and treatment, combining established caloric intake-reducing therapies, such as GLP-1 receptor agonists, with interventions aimed at increasing energy expenditure via activation of p38α and AMPK signaling pathways.

## 1. Introduction

### 1.1. Glucose Homeostasis and Type 2 Diabetes

Homeostatic glucose levels are maintained between meals by insulin, a hormone secreted by the pancreas that promotes glucose uptake by various cells throughout the body for energy production, thereby stabilizing blood sugar levels. Type 2 diabetes mellitus (T2D) is a chronic metabolic condition characterized by persistent hyperglycemia, resulting from a combination of insulin resistance, primarily in muscle and adipose tissues, and insufficient insulin secretion from pancreatic beta cells. The onset of type 2 diabetes is largely associated with obesity and a sedentary lifestyle, although some individuals may have a genetic predisposition that increases their risk [1]. Obesity and a sedentary lifestyle are major contributors to the loss of insulin sensitivity and reduced glucose uptake in peripheral tissues, particularly skeletal muscle, liver, and adipose tissue [2,3].

### 1.2. Lipid-Induced Insulin Resistance in Peripheral Tissues: Two Prevailing Models

Under normal physiological conditions, insulin facilitates glucose uptake by activating the canonical insulin receptor (IR)–insulin receptor substrate (IRS)–phosphoinositide 3-kinase (PI3K)–Akt signaling pathway (Figure 1A). This process includes the phosphorylation and inactivation of Akt substrate 160 (AS160), a protein that, when activated, inhibits the translocation of glucose transporter 4 (GLUT4) to the cell membrane. Therefore, the Akt-mediated inhibition of AS160 encourages the fusion of GLUT4-containing vesicles with the plasma membrane, leading to an increase in glucose uptake [4]. Insulin resistance is a state in which peripheral tissues, particularly muscle, fat, and liver, fail to respond adequately to insulin, resulting in impaired GLUT4-mediated glucose transport due to defective translocation of GLUT4 to the cell surface. While insulin resistance primarily impacts carbohydrate metabolism, the main contributors to this condition are lipids, particularly saturated fatty acids [5].

Over the years, two prominent hypotheses have emerged to explain the role of lipids in the development of insulin resistance and glucose intolerance (Figure 1). One model relates to the inhibitory effect of accumulated intracellular lipid derivatives inhibiting components of insulin signaling [6]. According to this model, lipid-signaling molecules such as diacylglycerols (DAGs) and ceramides, which are synthesized in the cytosol from accumulated acylated fatty acids, activate specific stress kinases, notably PKCθ and PKCξ. These kinases inhibit key insulin-signaling molecules, including insulin receptor substrate (IRS) and Akt. Since insulin signaling is crucial for orchestrating the trafficking of glucose transporters (GLUTs) to the plasma membrane and facilitating glucose uptake into cells, its inhibition disrupts the initial phase of glucose transport (Figure 1A). Another model, proposed by Randle and colleagues, introduced the concept of the “glucose-fatty acid cycle” [7] (Figure 1B). This model suggests that excess fatty acid supply leads to saturation of mitochondrial oxidation, causing accumulation of metabolites such as acetyl CoA, NADH, citrate, and ATP. These metabolites inhibit pyruvate dehydrogenase (PDH) and suppress glycolytic enzymes like phosphofructokinase (PFK) and hexokinase (HK), impairing carbohydrate oxidation through both insulin-dependent and -independent mechanisms. This model was supported by recent metabolomics studies that demonstrated a link between insulin resistance and the overload of mitochondria with lipids. Under these conditions, incomplete fat oxidation increases, leading to the buildup of long- and medium-chain acylcarnitines, contributing to ‘metabolic inflexibility’—the impaired ability to switch from fat to carbohydrate oxidation in response to feeding. This mitochondrial-centric model links these changes to glucose intolerance rather than directly to insulin resistance [8,9,10,11].

Both models highlight intramuscular lipid accumulation and mitochondrial dysfunction as contributors to insulin resistance. While cytosolic lipid buildup and incomplete β-oxidation may play roles, recent studies show that long-chain acylcarnitine accumulation in muscle is not associated with insulin resistance, indicating the need for further research to identify the specific lipid metabolites involved [12].

## 2. P38 MAPK in the Regulation of Glucose Homeostasis and Insulin Sensitivity

### 2.1. General

p38 kinases are proline-directed serine/threonine kinases belonging to the mitogen-activated protein kinase (MAPK) family. These kinases are present in all eukaryotic organisms, and their structural and regulatory features are conserved from yeast to humans. Unlike prototypic MAPKs, p38 and c-Jun N-terminal kinase (JNK) do not typically respond to mitogens; instead, they are activated by a wide range of stresses, including environmental and intracellular insults, as well as pathologies such as infection and tumorigenesis. Consequently, this group of MAPKs is classified as stress-activated protein kinases. However, p38 MAPK and JNK exhibit different and occasionally conflicting functions. In the context of this review, JNK has been suggested to contribute to insulin resistance [13,14,15], whereas p38 MAPK is more often implicated in positive regulation of energy expenditure [16]. The p38 MAPK family includes four members p38α, p38β, p38γ and p38δ [17]. In this review, we mainly discuss p38α but also mention the roles of other family members in energy metabolism. Given the diverse processes that p38 kinases may regulate, dysregulation of this pathway has been associated with several diseases, indicating that pharmacological targeting of p38 signaling could hold therapeutic potential [18].

### 2.2. Whole Body Involvement of p38α MAPK in Energy Metabolism Affecting Insulin Sensitivity

p38α MAPK has been implicated in the regulation of cellular bioenergetics across various levels and tissues, as thoroughly summarized in other reviews [18,19]. To facilitate a later discussion on potential targeted therapies aimed at p38α, we emphasize its role in energy metabolism and its impact on glucose homeostasis and insulin sensitivity in key tissues, including adipocytes, the liver, and pancreatic β cells. (Its involvement in skeletal muscle metabolism is later discussed in more detail).

In general, p38α primarily influences energy metabolism through its phosphorylation of transcription factors such as MEF2, CREB, ATF2, PPARα-δ, and C/EBPβ, and the co-activator peroxisome proliferator-activated receptor γ coactivator 1-α (PGC-1α) [19,20]. Many of the metabolic activities of p38α are mediated by the pleiotropic co-activator PGC-1α, which regulates energy metabolism across all tissues. p38α enhances PGC-1α activity through at least two mechanisms: it induces the transcription of the *pgc1α* gene [21] and it directly phosphorylates the protein, leading to the release of its repressor, 160MBP, and stabilizing the protein [22].

#### 2.2.1. Adipocytes

While p38α is well known for its roles in muscle and bone differentiation, its function in adipocyte differentiation is less clearly defined [23]. However, it plays a critical role in the thermogenic program of brown adipocytes and in the browning of white adipose tissue (WAT), largely by inducing Ucp1 expression [24]. This occurs via the PKA–ASK1–p38α signaling axis, which activates transcription factors such as ATF2, PPARγ, and PGC-1α to promote thermogenic gene expression [24,25]. The resulting uncoupling of mitochondrial respiration from ATP synthesis leads to generation of heat, a calorie-burning process stimulated by cold exposure that helps prevent weight gain. A recent study supporting the role of p38α in thermogenesis showed that IL-27 directly targets adipocytes, activating the p38 MAPK–PGC-1α signaling pathway and promoting UCP1 expression [26]. Interestingly, p38α is involved in beige adipocyte differentiation. IL-13 priming in preadipocytes drove beige adipogenesis in STAT6 and p38 MAPK dependency [27]. Surprisingly, mice lacking p38α in adipose tissue are resistant to diet-induced obesity and exhibit increased energy expenditure [28,29]. This paradox may stem from depot-specific regulatory differences or compensatory activation of other p38 isoforms, such as p38δ, which has been suggested to promote thermogenesis when not inhibited by p38α [28]. Additionally, p38α inhibition can restore browning capacity in aging by preventing progenitor cell senescence. These findings suggest that p38α plays context-dependent roles in adipose tissue, varying with cell type and physiological state, as outlined in recent reviews [18,30]

#### 2.2.2. Liver

p38 MAPK promotes hepatic gluconeogenesis by inducing key genes such as phosphoenolpyruvate carboxykinase (PEPCK) and glucose-6-phosphatase (G6Pase), primarily through the activation of PGC-1α [31,32,33]. This effect is partly due to p38α’s inhibition of the energy sensor, AMP-activated protein kinase (AMPK), which normally suppresses gluconeogenesis [34]. More recently, it was shown that p38α MAPK, in response to glucagon stimulation, promotes gluconeogenesis through the phosphorylation of FOXO1 at serine 273 [35].

Beyond glucose metabolism, p38 MAPK also plays a key role in liver lipid metabolism. It inhibits hepatic lipogenesis, as indicated by increased plasma triglycerides and liver fat when p38 MAPK activity is blocked [36]. This inhibition is associated with upregulation of *SREBP-1c* and its coactivator *PGC-1β*. Thus, p38 MAPK suppresses lipogenesis while promoting gluconeogenesis, a dual role that helps preserve glucose production during fasting by limiting lipid storage and conserving substrates for glucose synthesis. Simultaneously to inhibiting lipogenesis, p38 MAPK promotes hepatic β-oxidation, activities that prevent liver fat accumulation [37]. Indeed, activation of p38 MAPK in the liver, achieved by knocking out the MAPK phosphatase-1 (*mkp-1*) or *mkp-2* genes, prevented fat accumulation and steatosis in the liver under HFD conditions [37,38]. Similarly, activating p38 via constitutively active MKK6 reduced ER stress, improved blood glucose, and enhanced liver function in obese diabetic mice [39]. These findings support a protective role for p38α in preventing non-alcoholic fatty liver disease (NAFLD). However, p38α appears to play a dual role depending on disease stage: it protects against fat accumulation in early NAFLD when weakly activated but contributes to steatohepatitis in advanced stages when activation is stronger [40]. The delayed phase of steatohepatitis is driven by macrophage p38α, which promotes M1 polarization and pro-inflammatory cytokine secretion [41]. Notably, liver-specific expression of an intrinsically active p38α allele alone can induce fatty liver, even in lean mice [42]. The conflicting findings, showing both protective and harmful roles, likely reflect differences in experimental models, timing, activation levels, and signal duration, highlighting the need for further research effort.

#### 2.2.3. Pancreatic β Cells

Multiple studies suggest that p38 MAPK contributes to β cell dysfunction during the progression of type 2 diabetes. In response to chronic hyperglycemia, β cells increase insulin secretion, which can induce ER and oxidative stress. These stress conditions activate p38 MAPK, which—along with other stress kinases—is thought to impair insulin secretion and promote β cell apoptosis over time [43].

Analyses of pancreatic β cells lines correlated chronic p38α/β activity with loss of glucose-stimulated insulin secretion, β cell failure, and apoptosis [43,44]. Animal studies have confirmed the role of p38α/β in promoting β cell failure under diabetic conditions. Inhibition of p38α/β in db/db mice reduced hyperglycemia by preventing β cell apoptosis [45]. Another study linked increased p38α activity during aging to higher expression of cell cycle inhibitors (p16^Ink4a^ and p19^Arf^), reduced islet cell proliferation, and impaired regeneration [46]. Thus, p38α/β negatively impacts β cell function and viability. Interestingly, another member of the p38 family, p38δ, has also been shown to negatively regulate insulin secretion through the phosphorylation and inhibition of polycystic kidney disease 1 protein (PKD1), a protein important for β cell survival and insulin release [47]. Mice lacking p38δ were protected from stress-induced β cell death and exhibited increased insulin secretion.

Overall, these findings suggest that various p38 isoforms adversely affect β cell survival and function, highlighting the need for further research into their roles in health and disease.

#### 2.2.4. Systemic Involvement of p38α in Energy Metabolism

p38α also influences systemic energy expenditure by regulating the expression and secretion of metabolic hormones and responding to hormonal signals. One such hormone is fibroblast growth factor 21 (FGF21), which is secreted by the liver, adipose tissue, and skeletal muscle under metabolic stress conditions such as fasting, nutrient deprivation, and high-fat diet. FGF21 acts as an energy stress signal, enhancing lipid and glucose metabolism through its effects on both peripheral tissues and the central nervous system (CNS) [48]. The global metabolic effects of FGF21 represent a potential treatment for diabetes and other metabolic diseases [49]. p38α plays a key role in regulating FGF21 transcription and secretion in multiple tissues. In adipocytes, GPR120-induced FGF21 expression depends on p38 MAPK activation [50]. Similarly, brain tanycytes, glial-like cells involved in energy balance, produce FGF21 in a p38α-dependent manner [51]. In mice, endurance exercise increases circulating FGF21 and promotes the shift from fast- to slow-twitch muscle fibers through a p38 MAPK–dependent mechanism [52]. Together, these findings highlight the central role of p38α in mediating the beneficial metabolic actions of FGF21 across tissues.

p38α also mediates the metabolic effects of certain myokines, such as adiponectin, the most abundant peptide secreted by adipocytes and myofibers [53]. Adiponectin enhances glucose uptake and fatty acid oxidation in myotubes by binding to AdipoR1/R2 receptors on muscle cells [54]. Intracellularly, both AMPK and p38 MAPK signaling are co-activated via the adaptor protein APPL1 [55,56]. As formerly described, both kinases cooperate in increasing glucose uptake, mitochondrial biogenesis and fatty acid oxidation in muscle cells [57].

Irisin, an exercise-induced myokine, supports energy balance by lowering body weight and improving glucose metabolism. It enhances glucose uptake in myoblasts through the p38 MAPK–PGC-1α pathway and promotes the browning of white adipose tissue by increasing UCP-1 expression via p38 MAPK and extracellular signal-regulated kinase (ERK) [58].

Two recently identified small molecules, 3-methyl-2-oxovaleric acid and 5-oxoproline, are secreted by beige adipocytes and act via the cAMP–PKA–p38 MAPK pathway to induce brown fat–like traits in white adipocytes and enhance mitochondrial oxidative metabolism in skeletal muscle. In mouse models of obesity and diabetes, they reduce adiposity, increase energy expenditure, and improve glucose and insulin homeostasis [59].

In sum, several cytokines coordinate a systemic interorgan network regulating energy expenditure via p38 MAPK signaling.

The multifaceted role of p38 MAPK in regulating energy metabolism across glucose-regulating tissues must be carefully considered when developing therapeutic strategies targeting p38 MAPK for the treatment of obesity and insulin resistance.

## 3. p38α: A Key Regulator of Mitochondrial Biogenesis, Lipid Metabolism and Energy Turnover in Skeletal Muscle (Figure 2)

### 3.1. Metabolic Adjustment of Skeletal Muscle to Energy Requirements

Skeletal muscle acts as a key tissue for the uptake of glucose and free fatty acids, playing an essential role in the regulation of overall glucose metabolism and energy homeostasis. It serves as the largest glycogen storage organ, with a capacity that is four times greater than that of the liver. In mammals, skeletal muscle constitutes 40% of total body mass and is responsible for 30% of the resting metabolic rate in adult humans [60]. Under insulin-stimulated conditions, this tissue can remove 70–90% of the glucose load from the bloodstream [61].

**Figure 2 cells-14-01277-f002:**
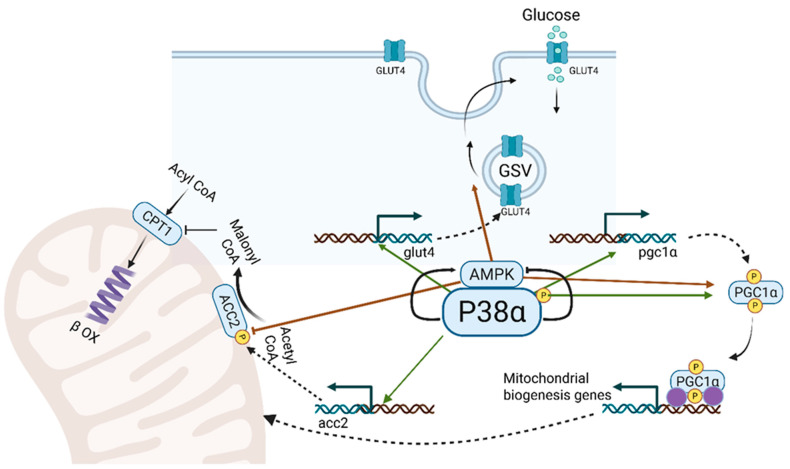
Metabolic activities of p38α MAPK and AMPK in the contracting muscle. p38α MAPK and AMPK cooperate in three key processes to enhance energy production: 1. Glucose Transport: p38α MAPK activates the transcription of the glut4 gene, producing GLUT4 protein, which is incorporated into GLUT4 storage vesicles (GSVs). AMPK facilitates the movement and fusion of GSVs with the sarcolemma, increasing GLUT4-mediated glucose uptake. 2. Mitochondrial Biogenesis: Increased energy demands during muscle contraction stimulate new mitochondria synthesis. p38α and AMPK enhance PGC1-α activity, the master regulator of mitochondrial gene transcription, by activating *pgc1α* gene transcription and phosphorylating PGC1-α. 3. Transport of Long-Chain Fatty Acids (LCFAs): Carnitine palmitoyltransferase I (CPT I) mediates the initial step in mitochondrial LCFA transport, negatively regulated by malonyl CoA from acetyl CoA carboxylase 2 (ACC2). p38α promotes *acc2* gene transcription, while AMPK phosphorylates and inhibits ACC2, regulating LCFA transport into mitochondria and β-oxidation according to energy needs of the contracting muscle.

Skeletal muscle is composed of various myofiber types, classified based on their contractile properties and metabolic profiles. Broadly, some fibers generate force rapidly but fatigue quickly, relying primarily on glycolysis for energy, while others contract more slowly but sustain force for longer durations, drawing energy mainly from mitochondrial oxidative metabolism [62].

A remarkable physiological feature of skeletal muscle is its capacity to swiftly adapt energy production, blood flow, and substrate utilization in response to physical activity. It is well established that increased contractile activity—such as during endurance exercise training—promotes a shift in muscle phenotype toward greater oxidative capacity. Endurance training induces metabolic remodeling, mitochondrial biogenesis, and angiogenesis, alongside other adaptive changes, ultimately enhancing insulin sensitivity and metabolic flexibility in both rodents and humans [63,64,65,66,67,68,69].

### 3.2. Metabolic Adaptations of Skeletal Muscle to Nutrient Excess and Energy Demand: Insights from Obesity and Endurance Exercise

Obese individuals accumulate intramyocellular triglycerides (IMTG), which is positively correlated with the degree of insulin resistance [70]. Surprisingly, endurance-trained athletes—despite being highly insulin sensitive—also exhibit elevated levels of IMTG compared to sedentary individuals. In fact, IMTG content in their muscles can exceed that observed in sedentary individuals with type 2 diabetes or obesity-related insulin resistance. This increase is thought to represent an adaptation to the elevated energy demands and enhanced lipid turnover of the active muscle. This seemingly paradoxical observation is known as the ‘athlete’s paradox.” [71].

In both endurance training and high-fat diet conditions, elevated IMTG levels result from increased delivery of circulating fatty acids to skeletal muscle by lipid transport proteins, including CD36, fatty acid binding proteins (FABPs), and cytosolic fatty acid shuttles. In both scenarios, fatty acid turnover is high, with increased mitochondrial import via carnitine palmitoyltransferase 1 (CPT1) [72] and entry into the β-oxidation pathway through the rate-limiting enzyme β-hydroxyacyl-CoA dehydrogenase (HAD) [73]. As a result, energy production in muscle relies heavily on fat oxidation, which in turn suppresses glucose oxidation by inhibiting pyruvate dehydrogenase (PDH), a key enzyme in the conversion of pyruvate to acetyl-CoA [74].

Therefore, the differences in insulin sensitivity observed in skeletal muscle during endurance training compared to conditions of excess high-fat diet (HFD) intake still need to be explained. The primary distinction lies in mitochondrial biogenesis, which is promoted by aerobic exercise but impaired in HFD-fed animals due to increased mitochondrial damage and insufficient compensatory biogenesis. As a result, mitochondrial oxidative capacity is enhanced in insulin-sensitive muscle and diminished in insulin-resistant muscle.

Endurance exercise is widely regarded as the most effective “treatment” for preventing, improving, and even reversing insulin resistance in peripheral tissues, particularly in skeletal muscle. p38 MAPK is vital for the adaptations required to fulfill the energy needs of contracting muscles, making it essential for maintaining glucose homeostasis. Muscle contraction triggers several intracellular changes, including calcium ion (Ca^2+^) influx, increased production of reactive oxygen species (ROS), heat, and mechanical stress, all of which contribute to the activation of the p38 MAPK pathway [75,76]. The degree of p38 MAPK activation is affected by exercise status [77,78]. During this process, p38 MAPK activates transcription factors such as ATF2, MEF2, and CREB which promote the expression of the *pgc1α* gene [21,79,80]. Constitutive activation of p38 MAPK enhances markers of mitochondrial biogenesis in skeletal muscle [21], and improves muscle strength [81] suggesting that this pathway is essential for mitochondrial adaptation to elevated energy demands. Finally, electrical stimulation in Duchenne Muscular Dystrophy model mice (mdx52) reduced muscle fatigue and promoted mitochondrial adaptations via the p38 MAPK/PGC-1α pathway, indicating its potential to improve muscle function even in severe degenerative disease [82].

### 3.3. p38α and γ in Mitochondrial Biogenesis

Mitochondrial biogenesis is a highly complex and regulated process that requires the coordination and co-expression of both nuclear and mitochondrial genomes for the assembly and expansion of the reticulum, as well as the generation of a dynamic mitochondrial network [83]. This complex process is regulated by the transcriptional co-activator, PGC-1α that orchestrates the expression of nuclear and mitochondrial genes essential for the contractile and metabolic adaptations of skeletal muscle [84,85]. Animal and cellular genetic models with altered expression of the PGC-1α gene have provided substantial evidence for its role in fiber type specificity [86], mitochondrial biogenesis [87,88], angiogenesis [89], GLUT4 expression [90], and improved exercise performance [91]. Yet, muscle-specific PGC-1α expression was not sufficient to affect insulin sensitivity, whereas exercise training did improve insulin sensitivity [92,93].

During muscle contraction, activated p38 MAPK enhances PGC-1α expression by phosphorylating its upstream transcription activators and increases its activity through direct phosphorylation (Figure 2) [16,94]. Although p38 MAPK’s role in regulating PGC-1α is well established, it remains unclear which isoform—α, γ, or both—is primarily responsible for this effect. Several murine exercise studies from the first decade of the 21st century that analyzed the p38 MAPK-PGC-1α axis in skeletal muscles were less specific about the isoforms involved, as they utilized genetically activated MKK3/6 transgenic mice [16,21,22,95]. Nevertheless, based on pharmacological selective inhibitors of p38α/β, such as SB202190, these studies suggested that p38α/β were involved in the activation of PGC-1α. Additionally, dominant-negative alleles of p38α or β, which carry two mutations in the activation loop of p38α or β and specifically attenuate the α and β isoforms without affecting the γ and δ isoforms, also indicated that p38α and β were involved in PGC-1α activation [21]. In vitro phosphorylation studies with purified p38α or β kinases demonstrated that p38α/β directly phosphorylated PGC-1α at sites that stabilized the protein and released PGC-1α from the 160 Myb repressor [22]. However, other studies identified p38γ as the activator of PGC-1α. In one study, overexpression of a dominant-negative form of p38γ MAPK, but not of p38α MAPK or p38β MAPK, blocked contraction-induced pgc1α transcription [96]. Another study indicated that p38γ MAPK is phosphorylated and activated in slow (soleus) but not in fast (gastrocnemius) muscles. Furthermore, the loss of p38γ MAPK reduced the number of slow myosin-expressing fibers and increased the number of fast myosin-expressing fibers in the soleus [97]. Finally, a recent proteomic study analyzing the metabolic parameters of muscles from transgenic mice and mouse primary myoblasts with loss and gain of different p38 isoforms reached the intriguing conclusion that PGC-1α activity and mitochondrial biogenesis were induced in the absence of the p38α gene, but not in its presence [81]. Can these differing conclusions be reconciled? Several studies, including our own, have shown that the activity of p38γ is significantly induced in models of muscle-specific p38α knockout [98,99]. Thus, it is formally possible that p38γ hyperactivity improved mitochondrial functions in the latter study [81]. Yet, the role of p38α in mitochondrial biogenesis in other animal models cannot be excluded. In a recent study, we observed elevated p38α activity in the insulin-resistant skeletal muscle of mice fed a high-fat diet (HFD), which contributed to maintaining mitochondrial integrity under metabolic stress induced by caloric excess [100]. Mice having attenuated p38α activity (p38α^AF^) [46] displayed impaired mitochondrial oxidative function and greater mitochondrial damage compared to controls. Although p38α alone is not sufficient to prevent insulin resistance in skeletal muscle under high-fat diet conditions, it helps protect against mitochondrial damage by promoting mitochondrial biogenesis.

In conclusion, accumulating results indicate that both isoforms α and γ are likely similar in their ability to activate PGC-1α in muscle tissue. Different metabolic stresses selectively activate p38α, p38γ, or both isoforms. For example, mechanical stress and elevated cytosolic Ca^2+^ during muscle contraction primarily activate p38γ, whereas the accumulation of lipid derivatives in the blood and muscle, due to a high-fat diet and sedentary behavior, mainly triggers p38α activation. Further studies should investigate whether different p38 isoforms phosphorylate the same or distinct sites on PGC-1α, and how these modifications influence its activity.

### 3.4. p38α and Lipid Oxidation

Despite the well-established roles of p38α in lipid metabolism within the liver, its function in skeletal muscle lipid metabolism remains less investigated.

Physical activity is critical in controlling the adaptation of fatty acid oxidation in skeletal muscles to the availability of lipid and glucose substrates [101]. In obese and sedentary individuals, muscle fails to effectively oxidize excess fatty acids through β-oxidation, and endurance exercise training is the most effective method to enhance fatty acid oxidation capacity in both obese and lean individuals [102]. AMPK is the key regulator of β-oxidation in mitochondria. AMPK is activated in the contracting muscle and by phosphorylating Acetyl CoA carboxylase 2 (ACC2) at Ser22, the activity of the latter is blocked. Inhibition of ACC2, a key enzyme in fatty acid synthesis, results in reduced levels of its product malonyl CoA, a potent inhibitor of CPT1 (carnitine palmitoyltransferase 1). CPT1 is an essential transporter of long chain fatty acids into the mitochondria for β-oxidation. The absence of malonyl CoA removes a barrier from CPT1 that enables maximal transport long-chain fatty acids (LCFAs) and β-oxidation [103]. However, exercise also induces alterations in fatty acid oxidation that are independent of the AMPK–ACC2–malonyl-CoA axis [104]. Therefore, there are also AMPK-independent pathways that modulate the adaptation of β-oxidation to meet the demands of contracting skeletal muscle.

The significance of p38α in the regulation of β-oxidation under conditions of excess fat caloric intake is proposed in our recent study [100]. Previous evidence for p38α’s involvement in lipid oxidation comes from studies showing its role in mitigating obesity in mice fed a high-fat diet (HFD) [37,100,105]. However, these earlier studies examined MKP-1 knockout mice, in which the activity of not only p38α but also JNK and ERK MAPKs was elevated. In a recent study, we compared the effects of an excess-caloric HFD on mice with attenuated p38α activity (p38α^AF^) relative to control mice. The p38α^AF^ mice exhibited greater weight gain and their skeletal muscle were more resistant to insulin compared to the control mice. Analyses of the tibialis muscles indicated accumulation of fat metabolites, mitochondrial damage and reduced β-oxidation in the muscles of p38α^AF^ mice relative to control mice. Further analysis of myotubes derived from both mouse strains led to the conclusion that p38α regulates two key processes involved in the selection of fat and carbohydrate oxidation in mitochondria. p38α regulates the mitochondrial transport of long-chain fatty acids (LCFAs) for oxidation by controlling the expression of the *acc2* gene. p38α also regulates the entry of pyruvate into the TCA cycle by facilitating the inhibitory phosphorylation of pyruvate dehydrogenase (PDH) at Ser 293 during β-oxidation. By regulating these two processes, nutrients are selectively oxidized in the mitochondria based on energy availability and demand, thereby supporting optimal mitochondrial function. In the absence of p38α activity, the selective regulation of mitochondrial carbohydrate and fat oxidation is impaired, potentially causing mitochondrial overload. This overflow in oxidative activity can lead to excessive production of reactive oxygen species (ROS), and metabolite byproducts resulting in damage to mitochondrial DNA and membranes [100]. Since this is the only study to date investigating the regulation of lipid oxidation by p38α in skeletal muscle, further independent studies using additional animal models are necessary to validate these findings.

To summarize, in the context of an excessive fat calorie diet, p38α coordinates two key events that are essential for preserving mitochondrial function and health (Figure 2). First, p38α promotes mitochondrial biogenesis, essential for enhancing oxidative capacity and preventing mitochondrial damage. Second, p38α regulates β-oxidation by controlling the transport of long-chain fatty acids (LCFAs) into mitochondria and modulates carbohydrate oxidation through its effect on pyruvate dehydrogenase (PDH) activity. Together, these coordinated functions of p38α help prevent mitochondrial overload from excess fatty acids and glucose, reducing the risk of damage from reactive oxygen species. Therefore, activating p38α in skeletal muscle may represent a promising therapeutic strategy to preserve mitochondrial function in obesity and insulin resistance.

### 3.5. AMPK Cooperates with P38 MAPK in the Metabolic Adaptation to Muscle (Figure 2)

AMP-activated protein kinase (AMPK), a serine/threonine kinase activated by metabolic stress and energy deprivation, is induced in contracting muscle [106,107,108]. It supports muscle adaptation by promoting mitochondrial biogenesis, β-oxidation, GLUT4 translocation, glucose uptake, glycolysis, and insulin sensitivity. Its transcriptional effects on oxidative metabolism depend on PGC-1α, with several regulatory mechanisms proposed and reviewed elsewhere [109,110].

P38 MAPK and AMPK are co-regulated in skeletal muscle responses to stress, energy metabolism, and exercise adaptation. Their co-activation under various metabolic stresses suggests functional interconnection, as both show synchronized activity during and after exercise in humans [111,112,113]. In conditions such as cardiac ischemia, AMPK can act upstream of p38 MAPK by recruiting it to the scaffold protein TAB1, thereby enabling its autophosphorylation [114,115]. Both kinases can also be activated in parallel by shared effectors [116,117]. For example, adiponectin co-activates AMPK and p38 MAPK signaling via the adaptor protein APPL1 [118]. In other metabolic stress conditions, AMPK and p38 MAPK activities are inversely correlated; for instance, in muscle denervation and high-fat diet models, elevated p38α activity suppresses AMPK activity [99,100]. Similarly, inactivation of the signaling kinase TAK1 (TGFβ-activated kinase 1), an inducer of skeletal muscle wasting, decreased p38 MAPK phosphorylation while increasing AMPK phosphorylation [119]. Thus, AMPK–p38 MAPK interplay in skeletal muscle is context-dependent, shaped by specific physiological or metabolic cues.

Intense interval exercise activates both AMPK and p38 MAPK, which converge on PGC-1α to enhance oxidative metabolism (Figure 2) [113,120]. These kinases synergistically activate PGC-1α at transcriptional and post-transcriptional levels. They also cooperate to promote glucose uptake: p38α increases *glut4* gene transcription, while AMPK facilitates GLUT4 vesicle translocation to the sarcolemma (Figure 2) [20]. Additionally, both regulate β-oxidation by modulating acetyl-CoA carboxylase 2 (ACC2); p38α drives *acc2* gene expression, whereas AMPK inhibits ACC2 activity via phosphorylation [100]. These activities of p38α and AMPK coordinate the mitochondrial transport of long-chain fatty acids (LCFAs) through carnitine palmitoyltransferase 1 (CPT1).

In summary, the close interrelations and complementary activities of p38α and AMPK in muscle energy metabolism should be considered in the development of therapies for insulin resistance in skeletal muscles.

## 4. Thoughts About Targeting p38α for the Prevention and Treatment of Type 2 Diabetes

The involvement of p38α in various responses of skeletal muscle to stress makes it a promising therapeutic target for various muscle-related diseases. The development of glucose intolerance and insulin resistance of muscle, fat and liver tissues is a key stage in the development of full-scale diabetes [121,122]. p38α is a critical for transport of glucose, to the selection of carbon sources oxidized in the mitochondria, and for the mitochondrial adjustment to the energy needs of the contracting muscle. Therefore, unlike the potential deleterious effects in excessive p38α activity in some pathological conditions, such as inflammatory diseases and cancer, its activity is necessary to optimize energy metabolism of skeletal muscle under stress. Thus, we suggest that a new strategy to combat obesity and treat type 2 diabetes is to develop approaches for increasing the activity of p38α in skeletal muscle. However, as discussed above, p38α activation in other organs, such as the liver and pancreas, can have adverse effects, including the promotion of hyperglycemia. Moreover, p38α/β activity contributes to muscle atrophy under pathological conditions such as denervation, unloading, and advanced-stage cancer [99,123,124,125]. Therefore, the success of this therapeutic strategy depends on selectively enhancing p38α activity in skeletal muscle while avoiding its activation in other tissues where it could contribute to disease progression.

Human behaviors that promote caloric balance remain the most effective strategies for preventing and improving the symptoms of type 2 diabetes. These behaviors include consumption of low-calorie diets combined with regular physical activity. The introduction of GLP-1-like molecules or agonists of the GLP-1 receptor, which primarily promote feelings of fullness and reduce food consumption, represents a significant advancement in diabetes prevention through weight loss. However, there is currently no pharmacological or molecular treatment that effectively mimics exercise by increasing skeletal muscle energy expenditure. Metformin (1,1-dimethylbiguanide hydrochloride), a long-established first-line medication for type 2 diabetes, primarily exerts its antidiabetic effects by inhibiting hepatic gluconeogenesis. Additionally, it may enhance glucose export and oxidative activity in skeletal muscles, likely through the activation of AMPK [126]. The concept of using pharmacological agonists of AMPK and PGC-1α as exercise mimetics to enhance muscle metabolism and prevent obesity-related diseases and type 2 diabetes in individuals unable to exercise was explored further by Evans and colleagues [127]. Their study demonstrated that oral targeting of AMPK could enhance exercise performance in mice. While AMPK agonists can replicate some metabolic effects of exercise, they cannot fully substitute for the comprehensive benefits of physical activity on metabolic health.

The combined actions of p38α in skeletal muscle, enhancing mitochondrial capacity and optimizing carbon substrate utilization, position it as a promising target for exercise-mimetic therapies. Furthermore, p38 MAPK’s role in mitochondrial uncoupling and thermogenesis in brown adipocytes, through the upregulation of uncoupling protein 1 (UCP1) expression, may also reduce mitochondrial ROS production and increase energy expenditure [24]. While it is unrealistic to expect that activating p38α in skeletal muscle—and possibly in adipose tissue—could fully replicate the broad benefits of physical training, it may enhance certain aspects of exercise metabolism.

A notable distinction between p38α and AMPK lies in their opposing effects on liver gluconeogenesis: while AMPK inhibits this process, p38α promotes it, leading to increased glucose release from the liver into the bloodstream [31,32,128]. Consequently, p38α agonists are anticipated to enhance glucose uptake and oxidation in the muscle while simultaneously increasing glucose release from the liver. In fact, these opposing actions of p38α in muscle and liver may contribute to improved glucose homeostasis and help prevent hypoglycemic events, which are common side effects of other antidiabetic medications. (e.g., sulfonylureas and, in rare cases, Metformin).

The catabolic activity of p38α, such as promoting energy expenditure and enhancing nutrient selection for oxidation, may be sufficient to induce weight loss. A study by Bennett and colleagues [105] demonstrated that mice with elevated p38/JNK activity due to the absence of MKP1 in skeletal muscles were resistant to high-fat diet (HFD)-induced obesity. Resistance to fat accumulation was not linked to variations in food intake or in increased physical activity of the above mouse strain. Instead, it suggests an increase in energy expenditure related to elevated levels of myofiber-associated factors. Similarly, we noticed a small but significant increase in weight gain in mice with attenuated p38α compared to control mice fed an HFD [100]. Thus, excessive p38α activity is expected to enhance the oxidative efficiency of skeletal muscle and increase their energy expenditure, particularly when combined with physical activity.

While AMPK activity is elevated in muscle following aerobic exercise [129,130], it is not increased by a high-fat diet [100]. In contrast, the activity of p38α is significantly elevated under both conditions, exercise and HFD [100,131,132]. The increased activity of p38α in response to a high-fat diet induces mitochondrial biogenesis and regulates the oxidation of excess intramuscular fat metabolites [100]. Yet, in the context of these conditions, p38 MAPK activity alone is inadequate to prevent mitochondrial damage and the onset of type 2 diabetes. The complementary catabolic activities of AMPK and p38α during and after endurance exercise enhance glucose export and improve mitochondrial oxidative capacity. Given that the activities of AMPK and p38α are not always co-activated but complementary, future treatments for type 2 diabetes should focus on developing strategies that can simultaneously activate both kinases.

## 5. Strategies to Augment p38α Pathway Activity (Figure 3)

The strategies proposed to enhance p38α activity are currently supported only by basic preclinical research. At this stage, they remain speculative approaches for treating obesity and type 2 diabetes in animal models. Advancement to human clinical trials would require prior confirmation of their efficacy and safety in appropriate preclinical systems.

**Figure 3 cells-14-01277-f003:**
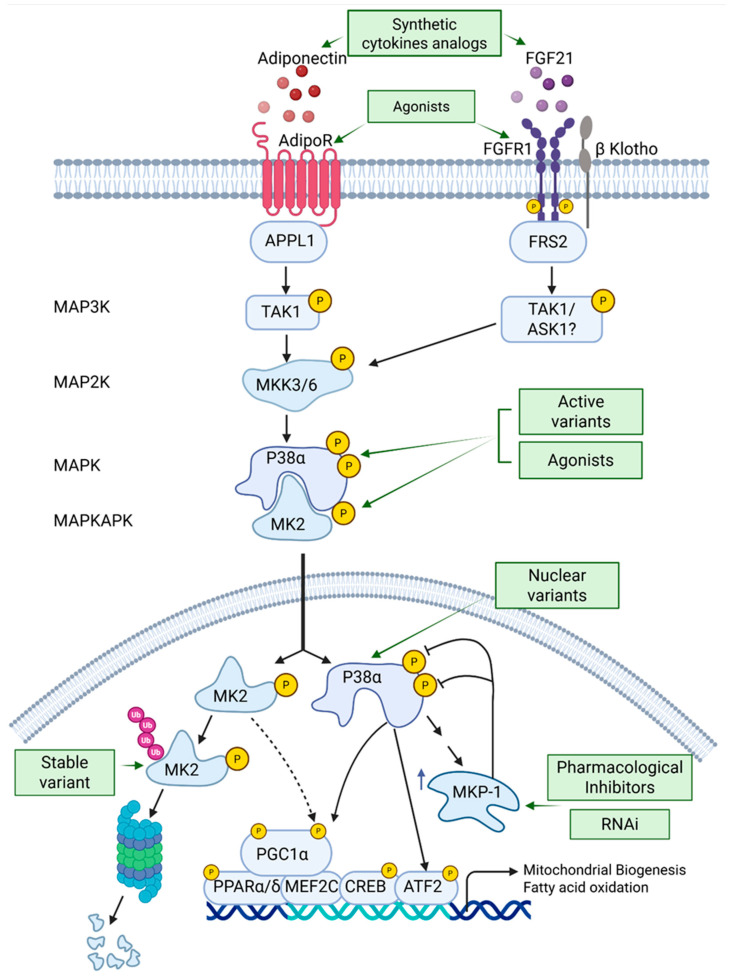
Activation of the p38 MAPK pathway as a potential treatment for insulin resistance and type 2 diabetes (T2D). The text outlines strategies to enhance the p38 MAPK pathway, either alone or with AMPK, including synthetic cytokine analogs, receptor agonists, active MAPK and MAPKAPK variants, and MKP-1 inhibitors. These approaches aim to increase and prolong p38 MAPK signaling. Abbreviations: FGF, fibroblast growth factor; APPL1, Adaptor protein, phosphotyrosine interacting with PH domain and leucine zipper 1; FRS2, Fibroblast growth factor receptor substrate 2; MKK3/6, MAP kinase kinase 6; MK2, MAP kinase-activated protein kinase 2; MKP-1, MAP kinase phosphatase 1.

### 5.1. Expression of p38α Variants in Skeletal Muscles

#### 5.1.1. Intrinsically Active Variants

Constitutively active variants of p38α have been developed and expressed in the liver hepatocytes of mice [42]. These mice exhibited macrovesicular lipidosis, resembling the phenotypic appearance of liver steatosis observed in obese individuals and patients with type 2 diabetes. This finding is surprising, as reduced p38α activity has previously been linked to increased hepatic fat deposition in conditions such as nonalcoholic fatty liver disease [133,134] while increased activity of p38 MAPK in livers of obese mice reduces endoplasmic reticulum (ER) stress and establishes euglycemia [39]. The activity of p38α in the liver likely mirrors its role in skeletal muscle, where it enhances mitochondrial oxidative capacity and regulates β-oxidation. A plausible explanation for the results observed by Engelberg and colleagues is the dramatic downregulation of p38α’s immediate downstream target, MAP kinase-activated protein kinase 2 (MK2) [42,135]. The induced expression of the active p38α variant rapidly led to MK2 degradation via the proteasome-ubiquitin system. Notably, MK2 degradation was also observed in other tissues upon the expression of the intrinsically active p38α (D.E., personal communication). MK2 protein levels were restored only after the expression of the active p38α variant was turned off. Therefore, the p38α-MK2 axis of p38 MAPK signaling is quickly inhibited by the active variant, which may partially explain the liver pathology reported in their study. For this reason, the continuous expression of hyperactive p38α variants in muscle fibers may not be a feasible therapeutic strategy for activating the pathway and alternative strategies for activating the pathway should be considered.

A likely explanation for the dramatic signaling shutdown observed with intrinsically active p38α variants is their amplified chronic activity, which significantly exceeds the physiological levels induced by wild type p38α. The excessive chronic activity of these p38α variants may be toxic to cells, prompting them to protect their viability through a substantial downregulation of the signaling pathway. In contrast, the expression of less active variant of p38α in skeletal muscles may not trigger such a downregulation, allowing for sustained signaling without compromising cellular health. The activity of the aforementioned p38 variants can be diminished through several strategies: (a) utilizing weak promoters to drive the expression of the p38α variant, (b) engineering variants that are prone to degradation by the proteasome by inserting E3-specific recognition sequence, and (c) selecting p38α variants that exhibit activity comparable to that of endogenously activated p38α.

#### 5.1.2. Expression of Nuclear Localized Variants

The role of p38 MAPK in muscle energy metabolism primarily involves the phosphorylation of transcription factors such as CREB, ATF2, PPARα, and C/EBPα, as well as co-activators like PGC-1α and YAP/TAZ [20]. Under normal cellular conditions, the majority of p38 MAPK protein resides in the cytoplasm, but it can translocate to the nucleus in response to various types of cellular stress [136]. Like other members of the MAPK family, p38α/β lack the typical nuclear localization signal (NLS) [137]. Movement of p38α/β into and out of the nucleus may be facilitated by other protein chaperones and their substrates [138]. Therefore, directing p38α into the nucleus by attaching an NLS to the kinase will increase its nuclear concentration and activity and promote the phosphorylation of key transcription mediators essential for its metabolic functions, such as PGC-1α.

### 5.2. Activation of p38α Targets Such as MK2

Another potential solution for activating the pathway is to focus on p38α targets, including MK2 and others. First, it is essential to investigate whether the p38α-MK2 axis is sufficient to mediate the beneficial metabolic effects of p38α in skeletal muscle. If MK2 can effectively substitute for p38α activity, then a strategy to directly activate MK2 independently of p38α could be a viable therapeutic approach. The development of active variants of MK2 or synthetic pharmacological agonists of MK2 may offer promising solutions for type 2 diabetes (T2D). Alternatively, identifying a degradation-resistant MK2 variant that remains stable after phosphorylation by p38α could potentially prolong the signaling duration of this axis.

### 5.3. Recombinant Adeno-Associated Virus (rAAV) as a Tool for Therapeutics Expression of Active Variants

An accepted approach for therapeutic expression of proteins is by engineering recombinant adeno-associated viral (rAAV) vectors with increased tropism for skeletal muscle of AAV1 or AAV6 serotypes. These serotypes preferentially infect myofibers enabling long-term persistent transgene expression while eliciting minimal immune responses, characteristics that make AAV vectors particularly well-suited for muscle-directed gene therapy [139]. In this context, the recombinant vectors would encode the active variants under the control of muscle-specific enhancers or promoters, such as α-actin or muscle creatine kinase (MCK) [140].

### 5.4. Inhibition of MAPK Phosphatase 1 (MKP1)

MKP1, also known as dual specificity phosphatase 1, is a member of the mitogen-activated protein kinase phosphatases (MKPs), which are a sub-family of dual-specificity phosphatases (DUSPs) within the larger superfamily of protein tyrosine phosphatases (PTPs) [141]. MKP1 specifically interacts with the active forms of p38, JNK, and to a lesser extent, ERK. This interaction activates MKP1, allowing it to dephosphorylate these stress kinases. This process facilitates transient stress signaling, which plays a crucial role in various cellular responses, including metabolic signaling, inflammatory responses, and cancer growth [142].

Research by Bennett and colleagues has established the involvement of MKP1 in regulating stress MAPKs, particularly p38 and JNK, during key catabolic events. Notably, their work revealed that the level of MKP1 protein is increased in the skeletal muscle of obese humans [105]. Their findings suggest that MKP1 is part of an important stress response mechanism that leads to reduced energy expenditure in skeletal muscle, thereby contributing to weight gain. This study is the first to link increased expression of MKP1 in the skeletal muscle of obese humans with concomitant dephosphorylation of p38 MAPK [105]. Furthermore, overexpression of MKP1 inhibits the expression and activity of PGC1α, a master regulator of mitochondrial biogenesis and energy expenditure, by impairing p38 MAPK-mediated phosphorylation of PGC1α. The knockout of MKP1 in skeletal muscles has been shown to render mice resistant to diet-induced obesity and insulin resistance. Equally significant is the finding that mice lacking MKP-1 in the liver (MKP1-LKO) exhibit resistance to hepatic steatosis [143]. Interestingly, mice that lack MKP-1 expression in skeletal muscle also demonstrate resistance to hepatic steatosis [105,144]. However, the precise molecular mechanisms underlying this muscle-liver relationship remain to be investigated.

Targeting MKP-1 as a potential strategy for the treatment of insulin resistance, type 2 diabetes (T2D), and obesity is therefore promising. Targeting of MKP-1 could be achieved by the following approaches:

#### 5.4.1. Pharmacological Inhibitors

Although MKP-1 has been extensively studied compared to other members of its enzyme family, the crystal structure of the human MKP-1 catalytic domain was only recently reported [145]. Inhibitors of MKP-1 have been identified through high-throughput screening efforts [146]. The high similarity of the PTP domain across MKPs presents a challenge in developing potent and specific MKP-1 inhibitors. However, with the crystal structure of MKP-1 is now available and structure-based drug design is feasible. MKP-1 expression is typically low in most unstressed tissues but is upregulated in obese states and type 2 diabetes, particularly in skeletal muscle. Muscle-specific expression increases the likelihood of successful MKP-1-selective inhibitor treatments with minimal side effects.

#### 5.4.2. Muscle-Specific Targeting of MKP-1 by Genetic Means

To effectively target skeletal muscle MKP-1 for the treatment of metabolic diseases, a dual promoter technology should be designed. This would involve a skeletal muscle-specific transcription system controlled by a human alpha skeletal actin promoter, utilizing antisense-based therapeutics against MKP-1 (RNA Interference of MKP-1). Recent studies have demonstrated the feasibility of tissue-specific oligonucleotide delivery using both viral and non-viral delivery vectors [147].

### 5.5. Treatment with Metabolic Hormones Analogs or Hormone Receptor Agonists That Activate p38α in Skeletal Muscle

The levels of various cytokines (exerkines) rise in the bloodstream during endurance exercise, playing a crucial role in regulating energy metabolism by affecting glucose uptake, lipid metabolism, and insulin sensitivity. Among these are adiponectin, FGF21, and irisin that are secreted mainly from skeletal muscle and adipose tissue and activate p38α in skeletal muscles. The activation of p38α by these hormones is associated with enhanced energy expenditure, reduced body mass, and improved insulin sensitivity. Consequently, various strategies have been investigated to boost the activity of these hormones as a potential therapeutic approach to enhance insulin sensitivity and glycemic control in patients with type 2 diabetes. One such effort includes the development of synthetic analogs of FGF21 designed to replicate its beneficial effects while improving its stability and bioavailability [148,149]. Similarly, researchers have investigated synthetic analogs of adiponectin and agonists of the adiponectin receptor, AdipoR1 and AdipoR2, that aim to replicate the positive effects of this hormone on metabolism [150,151,152]. As of now, there are no known analogs or agonists specifically targeting irisin signaling that have been widely adopted or approved for clinical use in the treatment of type 2 diabetes (T2D). However, there is increasing interest in developing therapies that can mimic or enhance the effects of irisin to improve metabolic health. Researchers are exploring various strategies to harness the potential of irisin, given its role in promoting energy expenditure and improving insulin sensitivity. While promising, these efforts are still in the early stages, and further studies are needed to establish effective irisin-based therapies for metabolic disorders.

A major benefit of the cytokine strategy is the simultaneous activation of p38 MAPK and AMPK in muscle cells, which work together to effectively promote all the processes required to increase insulin sensitivity. Co-activation of the two kinases was effectively illustrated through the adiponectin–AdipoR–APPL1 pathway [56]. One important caution to consider with this strategy is the potential of obese people to develop resistance to FGF21, indicating a reduced ability to respond to FGF21 [153,154]. While serum levels of FGF21 are elevated in individuals with obesity, intracellular signaling is partially inhibited, primarily due to the downregulation of FGFR1 and its co-receptor β-klotho. Therefore, persistent high levels of FGF21 may be ineffective, and in such scenarios, strategies should focus on restoring sensitivity to FGF21.

## 6. Conclusions

Increased p38α activity in skeletal muscles under pathological conditions such as obesity and type 2 diabetes, as well as in healthy individuals engaged in endurance exercise, has led to confusion among researchers regarding whether the role of p38α is detrimental or advantageous [20]. Recent studies have concluded that while p38α activity in mice on a hypercaloric high-fat diet enhances certain beneficial metabolic functions, it is not sufficient to prevent insulin resistance in skeletal muscles [100,105]. Consequently, the prevailing view is that p38 isoforms, primarily α and γ, play a key role in optimizing glucose and fat oxidation and enhancing mitochondrial oxidative capacity in skeletal muscle under both healthy and pathological conditions.

The optimal physiological condition for maintaining mitochondrial health and maximizing oxidation of nutrients by muscles is attained through endurance exercise, which remains the most effective approach for preventing or managing type 2 diabetes. Endurance exercise enhances the activity of p38α/γ in skeletal muscles, and recent studies suggest that this increase promotes fat and sugar oxidation in a manner that could improve insulin sensitivity [52,96,112]. However, there are currently no suggested therapies aimed at modulating p38 MAPK activity. Most of the efforts by pharmaceutical companies have focused on developing inhibitors of p38 MAPK isoforms for the treatment of inflammatory diseases and cancer [155].

In this review, we have outlined the multi-organ metabolic activities of p38α, some of which could potentially help prevent conditions leading to the development of type 2 diabetes. Our primary focus is on the role of p38α in the metabolic stress of skeletal muscles induced by excessive caloric fat intake, during which p38α activity enhances glycolytic and mitochondrial oxidative functions. Additionally, p38α is essential for optimizing nutrient selection by the mitochondria, particularly in choosing between fat and sugar oxidation. We also examine the benefits and potential drawbacks of p38α activity in other tissues, including the liver and adipose tissue. This complexity is exemplified in the liver, where p38α activity inhibits lipogenesis and promotes β-oxidation, actions that help prevent fat accumulation and steatosis, while simultaneously enhancing gluconeogenesis and increasing glucose release into the bloodstream.

The metabolic activities of AMPK and p38α in skeletal muscle converge on similar processes. For example, both activities are induced in muscles during endurance exercise and cooperate in increasing glucose transport, glycolysis, and mitochondrial oxidation. For many years, metformin has been the first-line medication for the treatment of type 2 diabetes, primarily thought to increase muscle oxidative activity through its indirect activation of AMPK. We propose that towards achieving a more substantial effect on muscle oxidative capacity, the activation of the second key kinase, p38α, should also be considered. In the concluding section of this review, we outline potential strategies to activate the p38α pathway, emphasizing that these should be implemented in combination with AMPK activation for optimal metabolic benefits. These include methodologies for expressing active variants of p38α itself or its substrate MK2, as well as analogs of cytokines such as FGF21 and adiponectin, and inhibitors of the p38 MAPK phosphatase, MKP-1. The possible detrimental effects of p38α in inflammatory diseases and different cancers, along with its harmful impact on pancreatic β cells and the liver, which disrupts glucose homeostasis, underscore the need to confine its activation specifically to skeletal muscle. As a result, systemic pharmacological activation of p38α is less viable. Instead, emerging strategies such as hormone analogs and targeted delivery using recombinant adeno-associated viral (rAAV) vectors offer more promising and tissue-specific therapeutic avenues.

Future therapies for type 2 diabetes (T2D) should adopt a more holistic approach that addresses multiple facets of metabolic balance in the body. The latest advancements in GLP-1 therapy, which effectively control caloric intake, should be complemented by strategies aimed at enhancing and optimizing caloric expenditure in skeletal muscle. This process is intricately regulated by the AMPK and p38 MAPK stress pathways. By integrating therapies that not only reduce caloric consumption but also promote metabolic activity and energy expenditure, we can create a more comprehensive treatment strategy for managing T2D and improving overall metabolic health.

## Figures and Tables

**Figure 1 cells-14-01277-f001:**
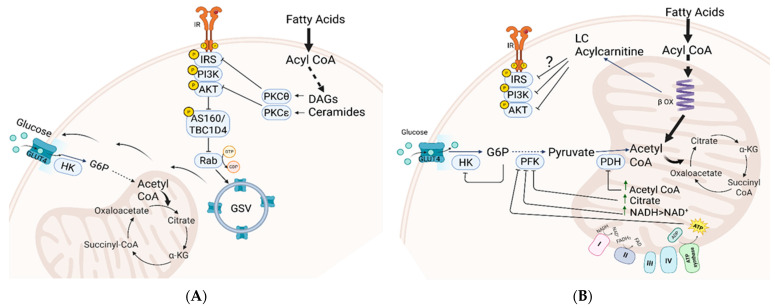
Two models illustrating lipid metabolism’s role in insulin resistance and glucose intolerance. (**A**) Accumulation of toxic lipid derivatives impairs insulin signaling. Lipid derivatives like diacylglycerols (DAGs) and ceramides activate stress kinases PKCθ and PKCξ, inhibiting insulin receptor substrate (IRS) and Akt. This disruption decreases the incorporation of glucose transporter (GLUT) into the plasma membrane, impairing insulin-mediated glucose transport. (**B**) The Randle “glucose-fatty acid cycle.” Excessive fatty acid oxidation in mitochondria inhibits key enzymes in glycolysis and glucose oxidation, creating “metabolic instability” that hinders glucose transport to tissues via insulin-independent and dependent pathways. Abbreviations: GLUT4, glucose transporter 4; IR, insulin receptor; IRS, insulin receptor substrate; PI3K, phosphatidylinositol 3 kinase; HK, Hexokinase; PFK, phosphofructokinase; PDH, pyruvate dehydrogenase; PKC, protein kinase C; DAG, diacylglycerol; GSV, Glut4 storage vesicles.

## Data Availability

No new data were created or analyzed in this study.

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
