# Peer review of "p38α MAPK Regulation of Energy Metabolism in Skeletal Muscle Offers a Therapeutic Path for Type 2 Diabetes"

_cells, 2025, doi:10.3390/cells14161277_

Round 1
Reviewer 1 Report
Comments and Suggestions for Authors
This review manuscript provides a comprehensive overview of the effects of p38α MAPK on skeletal muscle metabolism and explores its potential as a therapeutic target for type 2 diabetes.
Major Comments:
- While the manuscript presents an extensive body of information, it suffers from significant redundancy. In particular, the relationships between p38α, PGC-1α, and AMPK are discussed repeatedly across multiple sections, leading to fragmentation of the narrative. The logical flow requires substantial reorganization. Furthermore, the review heavily emphasizes the authors’ own research findings, raising concerns about objectivity. At this stage, where external validation and reproducibility remain limited, such an approach may mislead readers. A more balanced and critical comparison of evidence, including findings from independent studies, is essential for a credible review.
- The paper describes the diverse roles of p38α in various tissues—liver, adipose tissue, skeletal muscle, and pancreatic β-cells. However, it is problematic to treat these opposing effects—for example, enhanced gluconeogenesis in the liver versus increased glucose uptake in muscle—as a unified therapeutic target. In particular, the assertion that “activation in muscle is beneficial, whereas activation in the liver is undesirable” oversimplifies the complexity of tissue-specific signaling and lacks practical feasibility. The therapeutic potential of p38α modulation remains highly speculative under these circumstances.
- The proposition that “p38α activation could act as an exercise mimetic” is an interesting hypothesis. However, this claim lacks sufficient support from in vivo studies, particularly in humans. The generalization of findings from a single mouse model is scientifically premature. A more cautious tone is warranted, and any such claims should be supported by multi-center studies, independent replications, and clinical data.
Minor Comments:
- Figure legends are overly detailed, contributing to visual clutter. Reducing redundant explanatory text would improve readability and enhance the clarity of the figures.
- Numerous abbreviations (e.g., GSV, CPT1, ACC2, FGF21) are used without definition upon first mention, which may hinder reader comprehension.
- The references are disproportionately concentrated in the early 2010s. The manuscript would benefit from incorporating recent literature, particularly reviews and clinical studies published from 2022 onward.
Author Response
We thank the Reviewer for the constructive comments. The Reviewer raised three major points which we have addressed below.
Comment 1: “While the manuscript presents an extensive body of information, it suffers from significant redundancy. In particular, the relationships between p38α, PGC-1α, and AMPK are discussed repeatedly across multiple sections, leading to fragmentation of the narrative. The logical flow requires substantial reorganization. Furthermore, the review heavily emphasizes the authors’ own research findings, raising concerns about objectivity. At this stage, where external validation and reproducibility remain limited, such an approach may mislead readers. A more balanced and critical comparison of evidence, including findings from independent studies, is essential for a credible review.”
Response 1: The comment regarding redundancy across sections and fragmentation of the narrative was also raised by Reviewer 2 and has been addressed through substantial revisions. Specifically, we significantly streamlined the manuscript by eliminating or condensing multiple sections, resulting in a 20% reduction in overall length.
The most substantial change involved the complete removal of Section 2 from the original manuscript, titled “Skeletal Muscle Energy Metabolism.” This section had reviewed key aspects of skeletal muscle metabolism—such as fiber types, plasticity, and metabolic adaptation to energy demand—as well as the regulation of fatty acid metabolism in both obese animals on a high-fat diet and subjects performing endurance exercise. It also included detailed discussion of PGC-1α and AMPK as central regulators of mitochondrial biogenesis and oxidative metabolism.
Relevant material from this section was either omitted or condensed and integrated into a reorganized section now titled “p38α: A Key Regulator of Mitochondrial Biogenesis, Lipid Metabolism, and Energy Turnover in Skeletal Muscle” (originally Section 4, now Section 3). This restructuring not only eliminated redundancies but also improved the logical flow of the manuscript.
Regarding the reviewer’s concern that the review may overemphasize our own recent findings, we respectfully disagree with the assertion that this compromises objectivity. We did include discussion of our recent study, “p38α MAPK Coordinates Mitochondrial Adaptation to Caloric Surplus in Skeletal Muscle” (IJMB, 2023), as it is, to our knowledge, the only publication specifically addressing the role of p38α in nutrient-driven mitochondrial substrate selection and metabolic flexibility under high-fat diet conditions.
Given the novelty of the findings and the lack of comparable published studies in this specific area, we believe it is both appropriate and necessary to highlight this work. However, to maintain balance and avoid excessive focus, we have significantly reduced the discussion of this study in the revised manuscript. Experimental details have been omitted, and only the key conclusions are briefly presented in two short segments: one in Section 3.2 (“p38α in Mitochondrial Biogenesis”), beginning with “p38α is not activated exclusively during muscle contraction,” and another in Section 3.3 (“p38α and Lipid Oxidation”), beginning with “Our recent study highlights the significance of p38α...” Additional references to the study have been limited to concise, one-sentence mentions under relevant thematic contexts.
We believe this revised approach preserves the objectivity of the review while appropriately acknowledging a novel and relevant contribution to the field.
Comment 2: The paper describes the diverse roles of p38α in various tissues—liver, adipose tissue, skeletal muscle, and pancreatic β-cells. However, it is problematic to treat these opposing effects—for example, enhanced gluconeogenesis in the liver versus increased glucose uptake in muscle—as a unified therapeutic target. In particular, the assertion that “activation in muscle is beneficial, whereas activation in the liver is undesirable” oversimplifies the complexity of tissue-specific signaling and lacks practical feasibility. The therapeutic potential of p38α modulation remains highly speculative under these circumstances.
Response 2: We appreciate the reviewer’s comment regarding the complexity of p38α activity across different tissues and its varying effects on glucose homeostasis and insulin sensitivity. We fully agree that, in certain tissues—such as the liver—p38 MAPK promotes hepatic gluconeogenesis while also enhancing mitochondrial oxidative metabolism of fatty acids. This duality is precisely why we chose to include a broader discussion of p38α’s metabolic roles in multiple tissues, including its systemic involvement in energy metabolism.
A holistic view of p38α signaling is essential when considering its therapeutic modulation in complex metabolic diseases such as obesity and type 2 diabetes (T2D). To reinforce this point, we have explicitly highlighted the tissue-specific complexity of p38α activity in key sections of the revised manuscript. For instance, at the end of Section 2.2 (“Whole-Body Involvement of p38α MAPK in Energy Metabolism Affecting Insulin Sensitivity”), we added the following sentence:
“The multifaceted role of p38 MAPK in regulating energy metabolism across glucose-regulating tissues must be carefully considered when developing therapeutic strategies targeting p38 MAPK for the treatment of obesity and insulin resistance.”
Similarly, in the Conclusions section, we now state:
“The possible detrimental effects of p38α in inflammatory diseases and various cancers, along with its harmful impact on pancreatic β cells and the liver, which disrupts glucose homeostasis, highlight the importance of restricting p38α activation to skeletal muscles alone.”
Regarding the reviewer’s note that “the therapeutic potential of p38α modulation remains highly speculative under these circumstances,” we fully acknowledge this point. Indeed, the purpose of the review is to propose novel, yet unproven, conceptual strategies that may pave the way for future therapeutic approaches to combat obesity and T2D—the most prevalent metabolic disorders in the Western world. While speculative at this stage, we hope that the ideas presented in this review will stimulate further investigation and validation by the broader scientific community.
Comment 3: The proposition that “p38α activation could act as an exercise mimetic” is an interesting hypothesis. However, this claim lacks sufficient support from in vivo studies, particularly in humans. The generalization of findings from a single mouse model is scientifically premature. A more cautious tone is warranted, and any such claims should be supported by multi-center studies, independent replications, and clinical data.
Response 3: We would like to clarify that the article does not claim p38α can function independently as an exercise mimetic. In fact, we specifically reference the study by Evans and colleagues (Narkar et al., 2008, Cell 134, 405–415), which demonstrated that AMPK and PPARδ agonists could replicate some metabolic effects of exercise, but were insufficient to fully substitute for the broad physiological benefits of physical activity.
Furthermore, a substantial body of literature supports the idea that p38 MAPK is co-activated alongside AMPK during exercise and muscle contraction in both humans and rodent models. Several studies illustrate this co-activation and link it to enhanced mitochondrial biogenesis, lipid oxidation, and glucose uptake. Examples include:
- Hargreaves & Spriet (2020), Nat Metab 2, 817–828
- Combes et al. (2015), Physiol Rep 3
- Ihsan et al. (2015), Am J Physiol Regul Integr Comp Physiol 309, R286–R294
- Gibala et al. (2009), J Appl Physiol 106, 929–934
In line with these findings, our review emphasizes that future therapeutic approaches for type 2 diabetes should aim to harness the complementary activities of AMPK and p38 MAPK. Rather than viewing p38α as a standalone therapeutic target, we advocate for developing strategies that co-activate these two signaling pathways to mimic key aspects of exercise-induced metabolic improvements.
We also underscore throughout the article that this field is still in its early stages. While preclinical data are promising, we stress that rigorous clinical studies will be essential to validate the therapeutic potential of p38α in the treatment of type 2 diabetes and metabolic diseases.
Minor comments:
Figure legends are overly detailed, contributing to visual clutter. Reducing redundant explanatory text would improve readability and enhance the clarity of the figures.
Response: In response to the reviewer’s comment, we significantly reduced the length and level of detail in the figure legends of the revised manuscript.
Numerous abbreviations (e.g., GSV, CPT1, ACC2, FGF21) are used without definition upon first mention, which may hinder reader comprehension.
Response: We carefully reviewed the manuscript and ensured that all abbreviations are defined upon their first appearance in the text.
The references are disproportionately concentrated in the early 2010s. The manuscript would benefit from incorporating recent literature, particularly reviews and clinical studies published from 2022 onward
Response: Many of the foundational studies on p38α activity in muscle physiology were conducted during those years; however, we have incorporated several significant studies published from 2020 onward, which are now included in the revised version of the review.
Reviewer 2 Report
Comments and Suggestions for Authors
This review examines the contrasting metabolic adaptations of skeletal muscle to these opposing conditions. It highlights the key signaling molecules involved, focusing on the stress kinase p38α MAPK in skeletal muscle adaptation to overnutrition and endurance exercise. Overall, this well-written review demonstrates strong conceptual coherence, rich mechanistic detail, and forward-looking speculation about therapeutic targeting. This article can provide practical knowledge to the readers in metabolic biology, exercise physiology, and therapeutic development. Some comments may help to improve this article.
- Figure 1 needs to be enlarged. Some words are tiny and fuzzy.
- This article is quite long. Some text contents repeat similar themes. The overlapping contents can be consolidated to improve clarity and conciseness.
- P38 MAPK has been indicated to play pathological roles in many diseases, according to substantial publications. Thus, at the end of this article, a risk analysis of pharmacological p38α activation should also be included.
Author Response
We thank the reviewer for the constructive comments. In the overall impression, the reviewer noted that 'this well-written review demonstrates strong conceptual coherence, rich mechanistic detail, and forward-looking speculation about therapeutic targeting.' The reviewer also provided three minor comments, which we address below:
Comment 1: Figure 1 needs to be enlarged. Some words are tiny and fuzzy.
Response 1: The items in Figure 1 that previously appeared small and unclear—specifically the metabolites involved in the CREB cycle and the complexes of oxidative phosphorylation—have been enlarged to improve readability. In addition, following the request of Reviewer 3, the order of panels A and B has been reversed to align with their order of appearance in the main text.
Comment 2: This article is quite long. Some text contents repeat similar themes. The overlapping contents can be consolidated to improve clarity and conciseness.
Response 2: The reviewer’s comment is entirely justified, and we have made substantial efforts to address it. In response, we significantly reduced the overall length of the manuscript by removing redundant content and streamlining the narrative. For instance, the Introduction was notably shortened, and the entire Section 2, 'Skeletal Muscle Energy Metabolism,' was removed due to its overlap with sections focusing on p38α’s role in muscle energy metabolism. Relevant parts of the removed section were selectively integrated into the sections discussing p38α in mitochondrial biogenesis and lipid metabolism. These revisions have improved the clarity and coherence of the manuscript, while reducing the word count by more than 20% in the current version.
Comment 3: P38 MAPK has been indicated to play pathological roles in many diseases, according to substantial publications. Thus, at the end of this article, a risk analysis of pharmacological p38α activation should also be included.
Response 3: We address the pathological implications of p38 MAPK activity at several points throughout the manuscript. Nonetheless, we acknowledge the reviewer’s important comment and have responded by adding the following paragraph to the concluding section of the article:
'The possible detrimental effects of p38α in inflammatory diseases and various cancers, along with its harmful impact on pancreatic β cells and the liver, which disrupt glucose homeostasis, underscore the need to restrict its activation specifically to skeletal muscle. Consequently, systemic pharmacological activation of p38α is unlikely to be a feasible therapeutic approach. Instead, emerging strategies such as hormone analogues and targeted delivery using recombinant adeno-associated viral (rAAV) vectors represent more promising and tissue-specific alternatives.'
We believe this addition further clarifies the therapeutic limitations and contextualizes the importance of tissue-specific approaches.
Reviewer 3 Report
Comments and Suggestions for Authors
The authors discussed the beneficial role of p38α MAPK in skeletal muscle in treating type 2 diabetes. Athlete's paradox and its mechanisms were first reviewed; it is an interesting part to show the function of skeletal muscle in energy metabolism. The authors then considered metabolic benefits of p38α MAPK in different tissues, like adipose tissues, liver, pancreatic β Cells, and especially skeletal muscles. The mechanisms of p38α MAPK in skeletal muscles involve in promoted mitochondrial biogenesis and lipid oxidation. Lastly, they discussed the feasible treatment strategies which augmenting p38α pathway activity to fight type 2 diabetes. This manuscript is well written and organized, there are several minor concerns.
- The paragraph “Two models for explaining the development of insulin resistance in peripheral tissues” is less related to the topic. It might be omitted from this manuscript. In Figure 1, two images may be in reversed order.
- The paragraph “Skeletal muscle fiber type, and the metabolic adjustment to energy requirements” could also be omitted from this manuscript. In the latter sections, there is absence of p38α MAPK expression and activity in different skeletal muscle fiber type in regulating energy metabolism.
- The paragraphs 3.2.1-3.2.4 should be compressed because they are not focus of the manuscript. And there is repetitive information included.
- In paragraph “Strategies to augment p38α pathway Activity”, the authors should claim that most of these strategies are not tested in animal models or human patients.
Author Response
We thank the reviewer for his/her constructive comments. We appreciate the positive feedback noting that 'the manuscript is well written and organized,' and we have carefully addressed the minor concerns raised.
Comment 1: The paragraph “Two models for explaining the development of insulin resistance in peripheral tissues” is less related to the topic. It might be omitted from this manuscript. In Figure 1, two images may be in reversed order.
Response 1: We believe that both models are essential for understanding the development of insulin resistance in peripheral tissues. In particular, they provide critical context for the role of intramuscular lipid accumulation and mitochondrial dysfunction in impairing insulin action. This background helps frame the importance of p38α in regulating mitochondrial glucose and lipid oxidation, including its influence on mitochondrial biogenesis and β-oxidation.
The “glucose–fatty acid cycle” proposed by Randle is especially relevant to later discussions on fatty acid turnover, mitochondrial import via carnitine palmitoyltransferase 1 (CPT1), and the coordinated regulation of β-oxidation by AMPK and p38α. Given the review’s focus on mitochondrial health and nutrient oxidation, we chose to retain both models while significantly reducing the length of the corresponding paragraph for conciseness.
In accordance with the reviewer’s suggestion, the order of the two panels in Figure 1 has been reversed to match their presentation in the text.
Comment 2: The paragraph “Skeletal muscle fiber type, and the metabolic adjustment to energy requirements” could also be omitted from this manuscript. In the latter sections, there is absence of p38α MAPK expression and activity in different skeletal muscle fiber type in regulating energy metabolism.
Response 2: The sub-section “Skeletal muscle fiber type...” indeed contained information that could be streamlined or removed. Accordingly, we have omitted the entire Section 2 (“Skeletal muscle energy metabolism”), as it was partially redundant with the later sections discussing the role of p38α in skeletal muscle metabolism—now covered in Section 3 of the revised manuscript (formerly Section 4).
Relevant and necessary content from the original Section 2 has been integrated into the updated Section 3 to preserve coherence and flow. The detailed discussion of muscle fiber types was removed, and only a brief explanation was retained to provide essential context:
“Skeletal muscle is composed of various myofiber types, classified based on their contractile properties and metabolic profiles. Broadly, some fibers generate force rapidly but fatigue quickly, relying primarily on glycolysis for energy, while others contract more slowly but sustain force for longer durations, drawing energy mainly from mitochondrial oxidative metabolism.”
We believe this change improves clarity and maintains focus on the main theme of the review.
Comment 3: The paragraphs 3.2.1-3.2.4 should be compressed because they are not focus of the manuscript. And there is repetitive information included.
Response 3: The original paragraphs 3.2.1–3.2.4 address the metabolic roles of p38α in key glucose-regulating tissues—liver, adipocytes, pancreatic β cells—and its systemic effects. We agree with the reviewer that these sections are not the primary focus of the article but are essential for contextualizing potential therapeutic strategies targeting p38α. Given the complex and sometimes detrimental effects of p38α activity in tissues such as the liver and pancreatic β cells, we have substantially condensed these paragraphs (now 2.2.1–2.2.4 in the revised manuscript) in line with the reviewer’s suggestion.
Comment 4: In paragraph “Strategies to augment p38α pathway Activity”, the authors should claim that most of these strategies are not tested in animal models or human patients.
Response 4: We thank the Reviewer for this important comment. In the first paragraph of the section “Strategies to Augment p38α Pathway Activity,” we have added the following statement:
“The strategies discussed to enhance p38α activity are primarily based on preclinical animal studies and represent potential therapeutic approaches for obesity and type 2 diabetes. However, their efficacy and safety must be rigorously validated in suitable animal models before consideration for clinical application in humans.”
Round 2
Reviewer 1 Report
Comments and Suggestions for Authors
Overall, the authors have responded appropriately to the previous comments, and the manuscript has improved in both clarity and balance. However, the relationship between p38α, PGC-1α, and AMPK is still described in multiple sections, and further consolidation would be beneficial. In addition, while the discussion on therapeutic targeting is now appropriately cautious, it should be explicitly noted that the concept remains at the preclinical stage with respect to clinical application. Incorporating a few more recent reviews or clinical studies would also help to further enhance the currency of the information.
Author Response
We thank the Reviewer for the constructive comments. In the opening sentence, the reviewer stated: “Overall, the authors have responded appropriately to the previous comments, and the manuscript has improved in both clarity and balance.”
However, the reviewer also expanded on some earlier points, as detailed below:
1} However, the relationship between p38α, PGC-1α, and AMPK is still described in multiple sections, and further consolidation would be beneficial.
Response 1: The relationship between p38α, PGC-1α, and AMPK was previously discussed in Sections 3.2 and 3.4. We removed the paragraph on AMPK from Section 3.2 and merged all relevant content into a revised Section 3.4, now titled “AMPK cooperates with p38 MAPK in the metabolic adaptation to muscle contraction.” This section exclusively addresses AMPK’s role under various metabolic stresses in skeletal muscle, emphasizing its coordinated regulation with p38 MAPK during contraction. It highlights their cooperation in glucose transport and mitochondrial oxidative functions that improve insulin sensitivity, as shown in Figure 2.
Aside from brief mentions in other sections, necessary to explain mechanisms reported in cited studies, discussion of the p38α–PGC-1α–AMPK relationship is now confined to Section 3.4.
2} In addition, while the discussion on therapeutic targeting is now appropriately cautious, it should be explicitly noted that the concept remains at the preclinical stage with respect to clinical application.
Response 2: To address the reviewer’s comment, we revised the opening of Section 5 (“Strategies to Augment p38α Pathway Activity,” Figure 3) as follows:
“The strategies proposed to enhance p38α activity are currently supported only by preclinical research. These remain speculative approaches for treating obesity and type 2 diabetes in animal models. Advancement to clinical trials will require thorough validation of efficacy and safety in appropriate preclinical systems.”
This revision clearly conveys that the therapeutic approaches are preliminary and must be tested in animal models before clinical consideration.
3} Incorporating a few more recent reviews or clinical studies would also help to further enhance the currency of the information.
Response 3: After an extensive literature search, we found no relevant clinical studies directly addressing the topic. However, we incorporated several recent reviews and important research articles to enhance the manuscript’s currency. These new references have been cited in the appropriate sections. Below is a list of key additions:
- Herrera-Melle et al. (2024) Acta Physiologica — Demonstrates p38α’s role in regulating muscle strength via PGC-1α in mice (Ref 81).
- Cicuendez et al. (2021) Molecular Metabolism — Reviews stress MAPK involvement in liver steatosis and hepatocellular carcinoma (Ref 41).
- Whitehead et al. (2021) Nature Communications — Discusses adipokines inducing browning of white adipose tissue and mitochondrial metabolism in muscle via cAMP-PKA-p38 MAPK signaling, highlighting systemic energy expenditure networks (Ref 59).
- Wang et al. (2021) Nature — Shows IL-27 activates p38 MAPK–PGC-1α pathway in adipocytes, promoting thermogenesis and UCP1 expression (Ref 26).
- da Silva Rosa et al. (2020) Physiological Reports — Reviews insulin resistance mechanisms and cross-talk among muscle, liver, and adipose tissue, emphasizing skeletal muscle’s role (Ref 3).
- Leiva et al. (2020) Frontiers in Endocrinology — Summarizes p38 family roles in adipose tissue physiology, including adipogenesis and browning (Ref 30).
- Yamauchi et al. (2025) FASEB Journal — Reports that electrical stimulation in Duchenne Muscular Dystrophy model mice improves muscle fatigue and mitochondrial adaptations via the p38 MAPK/PGC-1α pathway, suggesting therapeutic potential in severe muscle degeneration (Ref 82).
These additions enhance the manuscript’s relevance by providing updated insights into p38 MAPK signaling in metabolism and disease. We hope our response satisfactorily addresses the Reviewer’s comments.